# Early Postnatal Neuroinflammation Produces Key Features of Diffuse Brain White Matter Injury in Rats

**DOI:** 10.3390/brainsci14100976

**Published:** 2024-09-27

**Authors:** John Waddell, Shuying Lin, Kathleen Carter, Tina Truong, May Hebert, Norma Ojeda, Lir-Wan Fan, Abhay Bhatt, Yi Pang

**Affiliations:** 1Division of Neonatology, Department of Pediatrics, University of Mississippi Medical Center, Jackson, MS 39216, USA; jpwaddell@umc.edu (J.W.); kcarter7@umc.edu (K.C.); nojeda@umc.edu (N.O.); lwfan@umc.edu (L.-W.F.); abhatt@umc.edu (A.B.); 2Department of Physical Therapy, School of Health-Related Professionals, University of Mississippi Medical Center, Jackson, MS 39216, USA; slin@umc.edu; 3Undergraduate Summer Research Program, University of Mississippi Medical Center, Jackson, MS 39216, USA; tctruong@go.olemiss.edu (T.T.);

**Keywords:** preterm birth, myelination, microglia, astrocyte, differentiation, neurogenesis

## Abstract

Background: Perinatal infection is a major risk factor for diffuse white matter injury (dWMI), which remains the most common form of neurological disability among very preterm infants. The disease primarily targets oligodendrocytes (OL) lineage cells in the white matter but also involves injury and/or dysmaturation of neurons of the gray matter. This study aimed to investigate whether neuroinflammation preferentially affects the cellular compositions of the white matter or gray matter. Method: Neuroinflammation was initiated by intracerebral administration of lipopolysaccharide (LPS) to rat pups at postnatal (P) day 5, and neurobiological and behavioral outcomes were assessed between P6 and P21. Results: LPS challenge rapidly activates microglia and astrocytes, which is associated with the inhibition of OL and neuron differentiation leading to myelination deficits. Specifically, neuroinflammation reduces the immature OLs but not progenitors and causes acute axonal injury (β-amyloid precursor protein immunopositivity) and impaired dendritic maturation (reduced MAP2+ neural fiber density) in the cortical area at P7. Neuroinflammation also reduces the expression of doublecortin in the hippocampus, suggesting compromise in neurogenesis. Utilizing a battery of behavioral assessments, we found that LPS-exposed animals exhibited deficits in sensorimotor, neuromuscular, and cognitive domains. Conclusion: Our overall results indicate that neuroinflammation alone in the early postnatal period can produce cardinal neuropathological features of dWMI.

## 1. Introduction

Over the past few decades, the survival rate of very premature infants (<32 weeks of gestation) has markedly increased. As a result, the clinical focus of prematurity birth-related complications has shifted from predominantly late preterm (32–37 weeks of gestation) to very preterm [1,2]. It is estimated that up to 50% of very preterm infants suffer from neurodevelopmental problems, which manifest across a wide range of disabilities and affect various neurological domains including sensory, motor, cognitive, attentional, and neurobehavioral functions [3,4,5,6]. Compared to the necrotic form of brain white matter injury (WMI), also known as periventricular leukomalacia (PVL), that has been the leading cause of neurological problems in preterm infants [7], the more subtle, diffuse WMI (dWMI) emerges as the predominant form nowadays [7,8]. The lack of specific therapies for dWMI underscores the urgent need to develop appropriate animal models to facilitate translational research. Although the etiology of dWMI is complex, maternal infection is one of the well-defined contributory factors [6,9]. At present, the underlying cellular, molecular, and neural circuitry mechanisms of neurological deficits within dWMI patients are largely unknown. Limited postmortem studies indicate that, in contrast to PVL, where oligodendrocyte (OL) progenitor cells (OPCs) are selectively damaged, dWMI primarily affects the differentiation of OLs [10]. For example, in their analysis of postmortem dWMI brain tissue, Buser et al. [11] found that the overall number of OL lineage cells remained unchanged, while the progression of OPCs into immature OLs was significantly hindered. In addition, emerging evidence suggests that axonal injury and dysmaturation of neurons might be the primary mechanisms underlying neurobehavioral deficits, underscoring the complexity of dWMI and highlighting the need for developing novel animal models, which is a significant challenge in studying dWMI.

We have previously established a PVL model by delivering lipopolysaccharide (LPS) intracerebrally to neonatal rats [12]. This model encapsulates key neuropathological features of human PVL, including neuroinflammation, OPC apoptosis, myelination deficits, white matter necrosis, and ventriculomegaly. In the current study, we hypothesized that low-grade neuroinflammation (using a significantly lower LPS dose compared to our PVL model) would primarily impact the development of OLs and neurons rather than causing direct cell damage, resulting in neurobehavioral impairments. This approach could provide valuable insights into the pathogenesis of dWMI and potential therapeutic strategies.

## 2. Materials and Methods

### 2.1. Reagents

All chemicals were obtained from MilliporeSigma (St. Louis, MO, USA) and Western blot reagents from Bio-Rad (Hercules, CA, USA). Antibodies were purchased from the following companies: myelin basic protein (MBP), Rip, glial fibrillary acidic protein (GFAP), postsynaptic density protein 95 (PSD95), synaptophysin, microtubule-associated protein 2 (MAP2): MilliporeSigma (Burlington, MA, USA); platelet-derived growth factor receptor-α (PDGFR-α): Santa Cruz Biotechnology (Dallas, TX, USA); ionized calcium-binding adaptor molecule 1 (Iba1): Wako Chemicals USA (Richmond, VA, USA); doublecortin (DCX): Cell Signaling Technology (Danvers, MA, USA); β-amyloid precursor protein (βApp): ThermoFisher Scientific (Waltham, MA, USA).

### 2.2. Animals and Treatments

This study was conducted following the National Institutes of Health Guide for the Care and Use of Laboratory Animals. The animal protocol was approved by the Institutional Animal Care and Use Committee at the University of Mississippi Medical Center. Time-pregnant Sprague Dawley rats from Envigo (Indianapolis, IN, USA) were kept in a controlled environment (12 h period of light/dark cycle), with free access to water and food. Rat pups were delivered naturally at the facility of the Center for Comparative Research. On postnatal (P) day 5, male and female pups were randomly assigned to a control group and LPS treatment group.

The protocol of inducing brain inflammation via intracerebral LPS injection was described in our previous study [12], with slight modifications about the coordinate of the injection site. The rat pup used in the experiment was anesthetized with isoflurane (4% for induction and 1.5% for maintenance). It was then positioned on a stereotaxic apparatus, with its head secured using a neonatal rat adapter (David Kopf, Tujunga, CA, USA). A midline incision was made on the scalp to expose the skull surface. Using a 10-μL precision syringe (World Precision Instruments, Sarasota, FL, USA) with an outer diameter of 0.20 mm, an intracerebral injection was administered at the following coordination: 1.0 mm posterior and 0.5 mm lateral to the bregma, and 2.5 mm deep into the skull. LPS (10 µg/kg) dissolved in saline was injected into the left hemisphere over 5 min. The injection volume was based on the rat’s body weight (1 µL per 10 g). The needle remained in place for an additional 5 min to allow LPS diffusion, then was slowly withdrawn. The wound was sutured, and the pups were placed on a thermal blanket (33–34 °C) for recovery, then returned to their dams. The control group received an equivalent saline injection. To reduce variations in body and brain size, pups from different litters were mixed and randomly reassigned to new litters, with each litter adjusted to 10 pups. According to our protocol, the injection targeted the corpus callosum just above the left lateral ventricle. A total of 82 neonatal rats were used in this study. The timeline for experiments is illustrated in Figure 1.

### 2.3. Immunohistochemistry

On P7 and P21, the rats were perfused transcardially with saline, followed by 4% paraformaldehyde (PFA). The brains were post-fixed in 4% PFA for 24 h and subjected to cryoprotection treatment (sequentially submerged in 10%, 20%, and 30% sucrose solution, 24 h each). Serial free-floating coronal sections (45 µm) were prepared using a freezing microtome (Leica, SM 2000R, Wetzlar, Germany) and stored at −20 °C. For immunofluorescence staining, the sections were rinsed extensively in PBS, blocked with 10% normal goat serum (Millipore, Billerica, MA, USA) in PBS containing 0.3% triton for 1 h at room temperature (RT), and incubated overnight at 4 °C with primary antibodies diluted in the blocking solution. The next day, the sections were washed three times with PBS and incubated with secondary antibodies conjugated with Alex Fluo488 (1:400) or 555 (1:2000) at RT for 1.5 h. After washing three times in PBS, the sections were mounted on slides and air-dried. Nuclei were counter-stained with DAPI (100 nM) dissolved in the Vectashield anti-fade mounting medium (Vector Laboratories, Burlingame, CA, USA). Images were acquired by a Nikon upright fluorescence microscope (Nikon NIE, Nikon Instruments Inc., Melville, NY, USA).

### 2.4. Stereological Cell Counting

Immunopositive cells were counted stereologically using the Stereo Investigator software package (Version SS-11, MBF Bioscience, Williston, VT, USA) [13]. A pilot experiment was performed to determine optimal counting parameters. Image stacks were acquired by a cooled monochrome camera that connected to the Nikon fluorescence microscope equipped with a motorized stage. Iba1+ microglia, platelet-derived growth factor receptor+ (PDGFR+) OPCs, and Rip+ immature OLs were counted in the corpus callosum of 4 coronal sections that were evenly spaced by 6 consecutive sections at the level of the frontal cortex. The area of interest was outlined under the DAPI channel, and a systemic random grid was virtually placed over the outlined contour. Image stacks (1 µm step size) were acquired using a 60× oil objective lens, and cells were counted using the optical fractionator probe. Considering the area of the corpus callosum is small, immunopositive cells were counted exhaustively using a relatively small systematical random sampling grid (160 µm × 120 µm) and a large counting frame (150 µm × 110 µm). The probe depth and guard zones were 18 µm and 2 µm, respectively. Total cell numbers were calculated by the software.

### 2.5. Neural Fiber Analysis by ImageJ

The area fraction of microtubule-associated protein 2 (MAP2)-immunostained neuronal dendrites was assessed by ImageJ software (Version 1.53q). For each animal, three consecutive coronal sections at the dorsal hippocampus were used for MAP2 immunofluorescence staining, and images were taken from six random sites in the cerebral cortex. For each site, image stacks (under 40× objective lens) were acquired first and then compressed into a single plan. For area fraction analysis of MAP2+ fibers by ImageJ, all images were thresholded at pre-determined parameters to highlight neural fibers. The area fraction was calculated as the ratio of the area occupied by MAP+ immunostaining fibers/total image area%.

Myelination was revealed by immunostaining of MBP at P21. The area fraction of MBP+ axons in the cingulate cortex was analyzed similarly to MAP2+ dendrites. For each animal, image stacks were captured at three randomly selected sites. The area fraction of myelination was calculated as: myelination index = (MBP+ fibers area/total image area)%.

### 2.6. Western Blot

The forebrain or hippocampus was dissected on ice under a stereomicroscope. To prepare whole-cell lysis, tissue was sonicated in cell lysis buffer containing 10 mM Tris, 100 mM NaCl, 1 mM EDTA, 1 mM EGTA, 1 mM NaF, 20 mM Na_4_P_2_O_7_, 2 mM Na_3_VO_4_, 1% Triton X-100, 10% glycerol, 0.1% SDS, 0.5% deoxycholate, 1 mM PMSF, and protease inhibitor cocktail (MilliporeSigma, St. Louis, MO, USA). The cell lysis was centrifuged at 12,000× *g* for 10 min at 4 °C. The supernatant was collected, and the total protein levels were determined using the BCA kit (ThermoFisher Scientific, Waltham, MA, USA). Samples were denatured and loaded to Bio-Rad TGX stain-free gels. After electrophoresis, proteins were transferred to a nitrocellulose membrane, blocked with 5% non-fat milk in PBS on a shaker for 1 h, and incubated with primary antibodies overnight at 4 °C. The member was washed 3 times in PBS and incubated with HRP-conjugated secondary antibodies at RT for 1 h. Signals were detected using the ECL select kit (Thermo Fisher Scientific), and images were acquired by the ChemiDoc MP Imaging system (Bio-Rad, Hercules, CA, USA). Data were analyzed by ImageLab software (Version 6.1.0 build 7, Bio-Rad, Hercules, CA, USA). Based on Bio-Rad’s stain-free technology, chemiluminescent signals of total proteins were used to normalize sample bands.

### 2.7. Behavioral Tests

A battery of behavioral tests assessing developmental milestones was conducted to evaluate neurological outcomes, based on previous work that used these tests to evaluate neurobehavioral toxicity [14,15]. Specifically, we used the righting reflex, negative geotaxis, wire hanging maneuver, and hind-limb suspension tests for assessing the development of neuromuscular functions in the early developmental stage (P6), and the vibrissa-elicited forelimb-placing and Y-maze tests for more complex functions reflecting cortical and subcortical maturation during later developmental stages (P21). An investigator who was unaware of experimental conditions conducted all tests.

#### 2.7.1. Righting Reflex

This test was used for assessing muscle strength and subcortical maturation [14,15]. Pups were placed on their backs, and the time taken to turn over onto all four feet and touch the platform was measured. Each pup underwent three trials on P8. With the time for each turnover recorded. The maximum allowed time for each trial was 60 s.

#### 2.7.2. Negative Geotaxis

This test evaluates the development of reflexes, motor abilities, and the integration of the vestibular labyrinth and cerebellum [15,16]. Rat pups were positioned on a 15° downward incline. Typically, healthy pups turn upward and start crawling up the slope. Each pup underwent three trials, and the time taken to turn 180° was recorded. We set the cut-off time at 60 s.

#### 2.7.3. Wire Hanging Maneuver

The test was designed to evaluate neuromuscular and locomotor development [14,15]. Pups were suspended by their forelimbs on a horizontal rod (5 × 5 mm^2^ area, 35 cm long, elevated 50 cm high by two poles). Typical healthy pups support themselves with their hind limbs to prevent failing and to aid in moving along the rod. A sawdust-filled box at the base provided protection in case the pups fell. Each pup underwent three trials, and the suspension latencies were recorded. The maximum time allowed for each trial was 120 s.

#### 2.7.4. Hind-Limb Suspension Test

This test evaluates the proximal hind-limb muscle strength, weakness, and fatigue in neonatal rats [17]. Pups were suspended by their hind limbs from the rim of a plastic cylinder (4 cm inner diameter and 16 cm in height), allowing them to hang off the rim. The suspension latency time was recorded from the moment the hind limbs were released until the pups fell into the tube, landing on a cotton ball cushion at the bottom for protection. The maximum time allowed for each trial was 120 s, and each pup underwent three trials.

#### 2.7.5. Vibrissa-Elicited Forelimb-Placing Test

This test was used to assess sensorimotor development in rodents [18]. The pup being tested was gently held by its torso and turned sideways so that its vibrissae were perpendicular to the table surface. The downward-facing limb was gently restrained, and the vibrissae were brushed against the table edge once per trial for a total of 10 trials. The percentage of successful trials in which the contralateral forepaw was placed on the tabletop was recorded for each side. Healthy rat pups typically place their forelimbs on the tabletop with 100% success in this test. Data from pups that struggled during the test were excluded from the overall analysis.

#### 2.7.6. Y-Maze

The Y-maze test evaluates spontaneous alternation behavior, which is indicative of attention and working memory [19,20]. The maze is constructed from black plexiglass and consists of three arms (50 cm long, 35 cm high, 12 cm wide) extending from a central platform at a 120° angle. The testing rat was placed at the end of one of the arms, allowing it to move freely among the three arms of the maze during the test session for 5 min. Spontaneous alternation behavior was defined as consecutive entries into all three arms. The sequence of arm entries was tracked using the ANY-Maze Video Tracking System (Stoelting Co., Wood Dale, IL, USA). The percentage of spontaneous alternation (the alternation score, %) for each rat was calculated as the ratio of the actual number of alternations to the maximum possible alternations (total arm entries minus two) multiplied by 100 (% alternation = [(number of alternations)/(total arm entries − 2)] × 100).

### 2.8. Data Analysis

Data were analyzed by a two-way ANOVA followed by a post-hoc Bonferroni test. If no sex differences were found, data from male and female animals within the same group were combined, and a two-tailed unpaired *t*-test was conducted. *p* < 0.05 was considered statistically significant.

## 3. Results

### 3.1. LPS Robustly Activates Microglia and Astrocytes in the Neonatal Rat Brain

We first sought to determine neuroinflammatory response by assessing the reactivity of microglia and astrocytes. Consistent with our previous report [12], intracerebral LPS exposure led to robust activation of microglia and astrocytes, as indicated by increased cell density and morphological characteristics 48 h post-challenge. Although Iba1+ microglia and GFAP+ astrocytes were activated across all brain regions, the hippocampus and periventricular white matter exhibited particularly heightened glial activation as seen by densely packed microglia and astrocytes with dramatic morphological changes (Figure 2), such as larger cell bodies, retracted processes, and brighter fluorescence staining. Stereology analysis revealed a more than 50% increase of Iba1+ cells in the corpus callosum region (Figure 3A). Due to their small cell bodies and long processes, it is technically challenging to count GFAP+ astrocytes accurately, so we determine GFAP contents in the whole forebrain by Western blot. Consistent with observations by GFAP immunostaining, the content of GFAP was 4.8-fold higher in the forebrain of LPS-exposed compared to the control rats at P7 (Figure 3B), and remained significantly elevated even at P21, suggesting a sustained glial reactivity during development.

### 3.2. Impairment in OL Maturation and Myelination without Loss of OPCs Following LPS Exposure

To assess the effect of neuroinflammation on OL maturation, we used lineage stage-specific markers to identify and quantify OPCs (PDGFR+) and immature OLs (Rip+) at P7. At this early postnatal age, PDGFR+ OPCs were found highly concentrated in the periventricular white matter tracks and, to a lesser extent, gray matter areas such as the overlying cortex and basal forebrain (Figure 4A). Stereological counting revealed that PDGFR+ cells were comparable between LPS and the control rats (*p* > 0.05), suggesting that neuroinflammation does not cause acute loss of OPCs. In contrast, the process-bearing Rip+ immature OLs were mainly observed in the major white matter tracks (Figure 4B). We found that, in contrast to OPCs, Rip+ immature OLs were reduced significantly in the periventricular white matter in the LPS-exposed rats (Figure 4D).

Given that myelination starts at the second postnatal week in rats, we assessed myelination by MBP immunostaining later at P21. As shown in Figure 5, MBP immunostaining in the major white matter tracks (e.g., corpus callosum, internal and external capsules, anterior commissure, etc.) was comparable between LPS and control groups. However, the density of myelinated fibers in the cingulate cortex was significantly reduced by LPS.

### 3.3. Neuroinflammation Leads to Acute Axonal Injury

The axonal injury was evaluated by double-immunofluorescence staining of βApp and phosphorylated neurofilament (pNF). βApp is normally present in the neuronal cell body but not axons, but is transported to stressed or degenerating axons; therefore, it can be used as a marker for axonal injury. As expected, we did not observe any βApp+ immunostaining in the control rat brains. However, brightly immunostained βApp+ neural fibers were observed in various brain regions of LPS-exposed rats, mostly noted in the white matter tracks (corpus callosum, internal and external capsules), but occasionally also found in gray matters, including the hippocampus and cortex (Figure 6B). To confirm these βApp+ fibers are indeed axons, double immunostaining of βApp+ with axonal marker pNF was conducted, which showed co-localizations (Figure 6A). Notably, some βApp+ long axons exhibited beaded morphology, a typical sign of acute axonal injury or degeneration.

### 3.4. Neuroinflammation Impedes Neurogenesis, Dendritic and Synaptic Maturation

To test whether neuroinflammation also affects neurogenesis and neural circuitry maturation, we assessed the expression of several markers expressed by neuroblasts, developing dendrites, and synapses. Western blot analysis shows that DCX expression in the hippocampus was significantly reduced in LPS-treated rats at P7 (Figure 7A). At P21, however, no difference in DCX was found between LPS and control groups (*p* > 0.05). Next, we performed immunostaining of MAP2, a specific marker for developing dendrites, to investigate whether the development of neural networks is also affected. As shown in Figure 7B, the cerebral cortex of P7 control rats developed extensive MAP2+ dendritic processes, while they were clearly fewer in LPS-treated rats. Area fraction analysis demonstrated more than a 3-fold reduction of MAP+ fibers. Alteration of dendritic development indicates that synaptic maturation is likely compromised by inflammatory insults. To test this, we further assessed two synaptic proteins, the postsynaptic density protein PSD95 and the vesicle protein synaptophysin, in the P21 rat brain hippocampus by Western blot. We found that PSD95 was reduced significantly by LPS exposure (Figure 7C), while synaptophysin was unchanged.

### 3.5. Neuroinflammation Leads to Functional Impairments

To assess functional outcomes of neuroinflammation, we conducted a battery of behavioral tests that evaluated the neuromuscular development.

#### 3.5.1. Righting Reflex

The mean latency times of LPS-exposed rat pups were significantly longer than the controls in both male and female rats (*p* < 0.001, Figure 8A). We did not observe sex differences in the righting reflex test.

#### 3.5.2. Negative Geotaxis

As shown in Figure 8B, the mean latency times of negative geotaxis in LPS-exposed male and female pups were significantly longer than the controls (*p* < 0.001). We did not observe sex differences in the negative geotaxis test.

#### 3.5.3. Hindlimb Suspension Test

This test evaluates the proximal hind-limb muscle strength, weakness, and fatigue in rat neonates 17. LPS-exposed pups showed much shorter mean latency times as compared to the controls (*p* < 0.005, Figure 8C). Although female pups tend to perform worse than male pups in this test, no statistical differences were detected.

#### 3.5.4. Wire Hanging Maneuver Test

Both male and female pups in the LPS group performed far worse than the controls. As shown in Figure 8D, LPS-exposed pups showed a dramatic reduction of mean latency times hanging on the rod than the controls (*p* < 0.001). No sex differences were observed in this test.

#### 3.5.5. Y-Maze Test

LPS-exposed rat pups scored a significantly lower success rate than the controls during the Y-maze test (*p* < 0.001). Furthermore, LPS-exposed males performed much worse than females (*p* < 0.001) (Figure 8E).

#### 3.5.6. Vibrissa-Elicited Forelimb-Placing Test

All control rat pups succeeded in the vibrissa-elicited forelimb-placing test, with a 100% success rate (Figure 8F). The success rates were significantly lower for both LPS-exposed male and female pups (*p* < 0.001) as compared to the control. Furthermore, LPS-exposed male pups performed far worse than female pups (*p* < 0.001).

## 4. Discussion

This study demonstrated that neonatal P5 rats (~32 wks of gestation in humans) exposed to neuroinflammation developed cardinal neuropathological features of dWMI, including dysmaturation of OLs and neurons, axonal injury, impaired myelination, and neurobehavioral impairments. Neuroinflammatory response, as assessed by microglia and astrocyte reactivity, persisted at least to juvenile age (P21), suggesting that once initiated, neuroinflammation could sustain for a long period and impose long-lasting adverse effects on the developing brain.

The most significant finding in this study is that a low- to moderate-grade neuroinflammation (as compared to a 10-fold higher dose of LPS used in a periventricular leukomalacia model [12]) led to arrest in the maturation rather than loss of OL lineage cells. Although PDGFRα is expressed in some neurons of the retina and Purkinje cells of the cerebellum, it is primarily expressed by OPCs in the forebrain and spinal cord and could be double-labeled with NG2 [21]. We observed that PDGFR+ OPCs were abundant in both the white and gray matter of LPS-exposed rats compared to the controls. However, process-bearing Rip+ immature OLs were markedly reduced 2 days following the LPS challenge, suggesting that the differentiation of OPCs towards immature OLs was blocked by inflammatory mediators, which were most likely generated from activated microglia and/or astrocytes. In a cell culture study, we demonstrated that conditioned medium from LPS-activated microglia blocks OPC differentiation [22,23], indicating a pivotal role of activated microglia in OL differentiation arrest and possibly myelination deficits under an inflammatory environment. The mechanisms of how activated microglia inhibit OPC differentiation are likely complex, given that homeostatic microglia play a two-faceted role in OL development and myelination during normal brain development. During normal development, microglia are intimately involved in supporting OPC survival, differentiation, and myelination [24] by releasing growth/trophic factors such as insulin-like growth factor-1 (IGF-1) [25]. Activated microglia, however, not only produce deleterious inflammatory mediators such as TNFα, IL1β, and nitric oxide but also lose beneficial effects mediated by growth/trophic factors. This idea was supported by our previous study demonstrating that either neutralizing TNFα or suppressing iNOS activity or supplementing exogenous IGF-1 or ciliary neurotrophic factor (CNTF) (both were markedly suppressed in microglia by LPS) significantly prevented OPC injury or differentiation arrest by LPS-activated microglia [20].

Reactive astrocytes, or astrogliosis, is frequently observed as a prominent hallmark of dWMI in post-mortem brain tissue [10,11]. Like microglia, astrocytes also express the LPS receptor toll-like receptor 4 (TLR4) and produce inflammatory mediators in response to LPS challenge [26]. Experimental studies demonstrated that reactive astrocytes adversely affect OL survival, maturation, and myelination by releasing cytokines [27,28], similar to activated microglia. In addition, reactive astrocytes produce high levels of hyaluronan, an extracellular matrix molecule that strongly inhibits OL differentiation and myelination [29]. Our finding that GFAP content was significantly elevated even at P21 (16 days after the LPS challenge) suggests sustained astrogliosis, which may have long-lasting adverse effects on the development of OL and likely neurons.

The subventricular zone (SVZ) and the dentate gyrus (DG) of the hippocampus are two major regions that continuously generate neurons postnatally. Newly generated neuroblasts or immature neurons that express specific markers DCX and NCAM migrate to the cerebral cortex or hippocampus and integrate into local circuitry [30]. We found that LPS-induced neuroinflammation significantly reduced DCX expression, suggesting that neuroinflammation may hamper postnatal neurogenesis. It is important to point out that reduction of DCX does not necessarily correlate with reduced neurogenesis, since DCX is a cytoskeleton protein involved in newborn neuron migration. To clarify this, we performed immunostaining to quantify Ki67+ cells in the SVZ and (DG). We found that Ki67+ cells, which are considered neural stem cells in the neurogenic niche of the postnatal rat brain, were significantly reduced in the DG but not SVZ (Appendix A).

Besides myelination deficits, axonal injury emerges as another potential mechanism contributing to neurological deficits in WMI infants [4,7]. Axonal injury may contribute to developmental deficits in gray matter, such as cortical and thalamic areas [4]. In addition to abnormal neurogenesis and axonal injury, we found that low-grade neuroinflammation also leads to abnormalities in dendritic and synaptic maturation, as indicated by reduced MAP2+ neural fiber density in the cortex. The postsynaptic density protein PSD95 plays a critical role in postsynaptic signaling transduction following activation of glutamate receptors [31], while the vesicle glycoprotein synaptophysin regulates the kinetics of synaptic vesicle trafficking [32]. We observed a significant decrease of PSD95 but not synaptophysin in LPS-exposed rats, which is consistent with the findings that PSD95 is reduced in a number of neurodevelopmental disorders such as Autism Spectrum Disorders (ASD) [33], and likewise, the findings that synaptic proteins are dysregulated in animal models involving neuroinflammation. For example, Lin et al. [34] found that systemic LPS exposure in neonatal rats caused a significant reduction of several synaptic proteins that was associated with the upregulation of proinflammatory mediators including TNFα, IL-1β, IL-6, and iNOS in activated microglia. It is worth mentioning that neurodevelopmental disorders, including WMI, ASD, and schizophrenia, have significant overlaps in etiology, including prenatal infection/inflammation as a common risk factor [35,36,37], while deficits in cognitive, attentional, and social behaviors are also common in WMI and ASD patients [38,39]. This suggests that there may be shared underlying neurobiological mechanisms among these mental disorders, especially during the early prenatal developmental period. Impairments in neurogenesis and neural network maturation are likely significant contributors to cognitive and behavioral deficits.

A large body of research suggests that the mechanisms underlying inflammation-mediated negative regulation on the development of neurons are similar to those of the OL lineage. Like OPCs, microglia-produced neurotrophic factors such as IGF-1 are also essential for the proper development of neurons. Thus, suppression of neurotrophic factors in activated microglia contributes to impaired neuronal development. Inflammatory cytokines are generally considered deleterious to neurogenesis. For example, systemic LPS challenge causes a significant decrease in DCX+ neurons in the rat hippocampus [40]. LPS also causes a significant reduction in parvalbumin+ (PV) interneurons in the hippocampus, which is mediated by microglial activation [41]. A large body of research suggests that PV+ interneuron dysregulation plays a role in neuropsychiatric disorders such as autism [42]. For instance, an imbalance of excitation to inhibition neurotransmission is the principal underlying molecular mechanism of seizure, which is common in neurodevelopmental disorders. Interestingly, neonatal mice exposed to the LPS challenge exhibit increased hippocampal excitatory synaptogenesis and enhanced seizure susceptibility, which is mediated by activation of TLR4 on astrocytes [43]. As mentioned earlier, neurodevelopmental disorders have substantial overlap in etiology, neurobiology, and symptoms. The finding that LPS-mediated neuroinflammation disrupted neurogenesis and synaptic maturation suggests that specific neurobehavioral deficits observed in dWMI patients may be principally caused by deficits in neuronal development.

We used various behavioral tests to assess the impact of neuroinflammation on different aspects of brain maturation. It is worth mentioning that sex differences were detected in two (vibrissa and Y-maze) of six behavioral tests, with male rats performing significantly worse than females. This finding is in agreement with clinical observations that preterm infants suffering from neurodevelopmental disorders also show sex differences, and in general, males are affected more severely than females [44,45]. The underlying neurobiological mechanisms of sex difference observed in our study are unknown; we speculate that it might be related to impairments in the development of higher-order functions such as cognitive function, as shown by the Y-maze test, or brain regions involved in higher-order functions such as the cingulate cortex. Interestingly, we did observe a male-specific impairment in myelination at P21 in the cingulate cortex, while myelination in other brain regions such as the major white matter tracks (e.g., corpus callosum, internal and external capsule) appears unaffected by LPS challenge.

A significant limitation of this study is the challenge of discerning whether the observed functional deficits are primarily due to developmental impairments in OL/myelination or neuronal components, such as alteration in neurogenesis, synaptic maturation, axonal injury, and network formation. Due to an intimate relationship between OL/myelin and neurons, both likely contribute to neurological dysfunctions reflected in various behavioral tests. For example, the myelin sheath is not only essential for the proper conduction of action potentials; it also plays a pivotal role in sustaining axonal health by supplying energy metabolism [46]. Likewise, the proper development of OLs and myelination also depends on instructional signals from neurons. Thus, compromise in the development of one cell type will ultimately affect the other and contribute to functional deficits indirectly. From a therapeutic standpoint, interventions targeting OLs/myelination will likely also benefit neurons, and vice versa.

## 5. Conclusions

Our study revealed several clinically relevant neuropathological features in a novel inflammation-driven dWMI rat model, which could be used as a model system for testing interventional therapeutics or investigating cellular/molecular mechanisms of dWMI.

## Figures and Tables

**Figure 1 brainsci-14-00976-f001:**
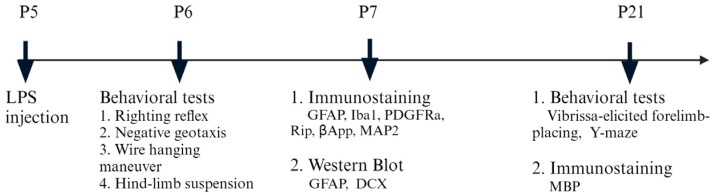
Experimental design and workflow.

**Figure 2 brainsci-14-00976-f002:**
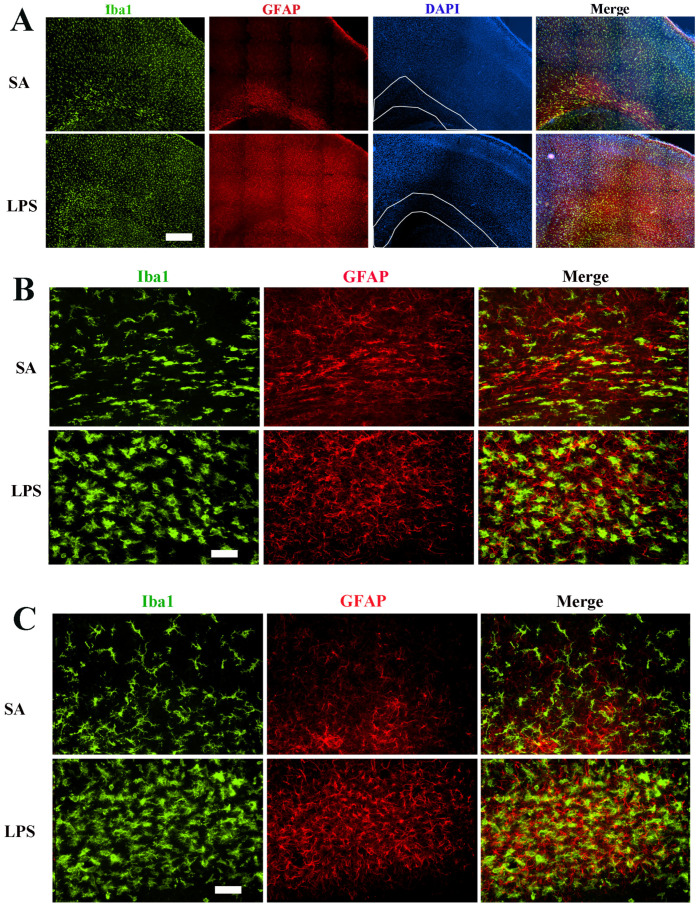
Robust activation of microglia and astrocytes 48 h following LPS exposure. (**A**) Representative low-power micrographs show a global increase in the density of Iba1+ microglia and GFAP+ astrocytes in the cerebral cortex and subcortical white matter (areas indicated by white lines). (**B**) Higher magnification images depict typical morphological characteristics of activated microglia and astrocytes in the subcortical white matter. LPS-activated microglia exhibit increased Iba1 immunofluorescence intensity, larger soma size, and retracted processes. (**C**) Extensive microglia and astrocytes in the LPS-exposed rats as shown in the hippocampus region. Scale bars: (**A**) (500 µm), (**B**,**C**) (50 µm).

**Figure 3 brainsci-14-00976-f003:**
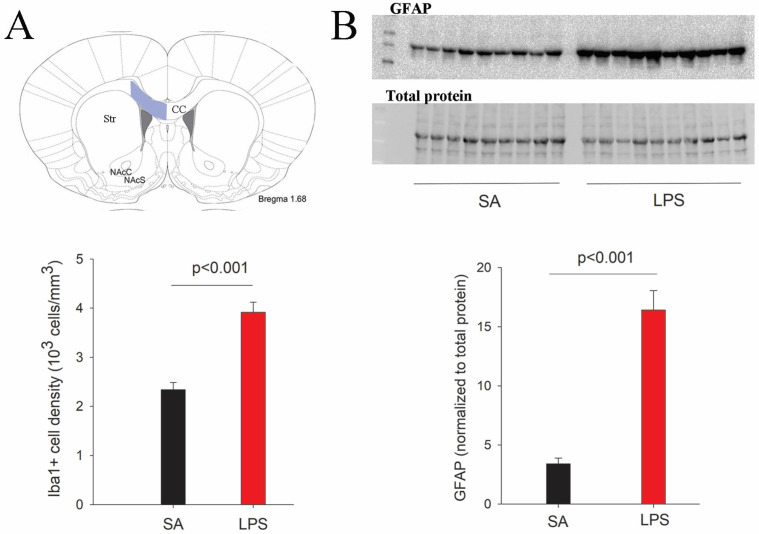
Quantification of microglia and astrocytes in LPS-exposed rats at P7. (**A**) Stereological counting of Iba1+ cells reveals a significant increase of microglial numbers in the corpus callosum area (highlighted in blue) following the LPS challenge. (**B**) Western blot indicates a more than 3-fold increase of GFAP content in the forebrain of LPS-exposed rats. *n* = 10 for each group.

**Figure 4 brainsci-14-00976-f004:**
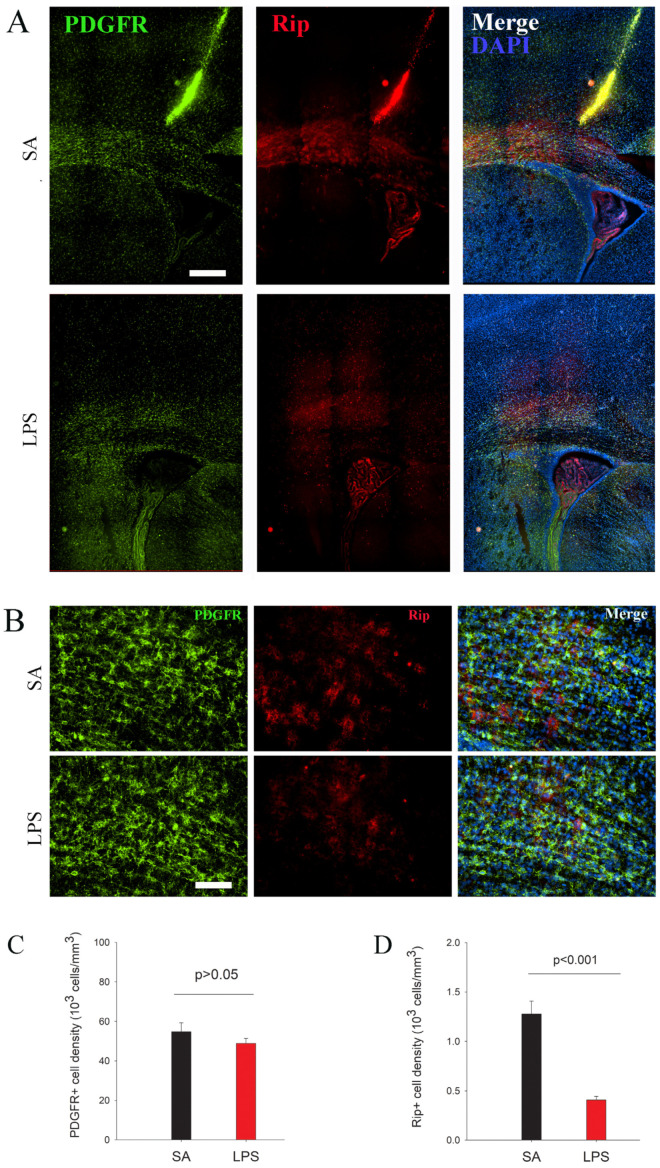
The arrest of differentiation rather than OPC loss following LPS exposure at P7. (**A**) Immunofluorescence staining reveals that PDGFR+ OPCs, most of which were observed in the subcortical white matter, were not affected by LPS exposure. However, the number of Rip+ immature OLs was significantly decreased in LPS-exposed rats. (**B**) depicts a high-power view of PDGFR and Rip immunostaining from subcortical white matter. (**C**,**D**) PDGFR+ and Rip+ cells were counted in the corpus callosum by stereology. *n* = 10 for each group. Scale bars: (**A**) (200 µm), (**B**) (50 µm).

**Figure 5 brainsci-14-00976-f005:**
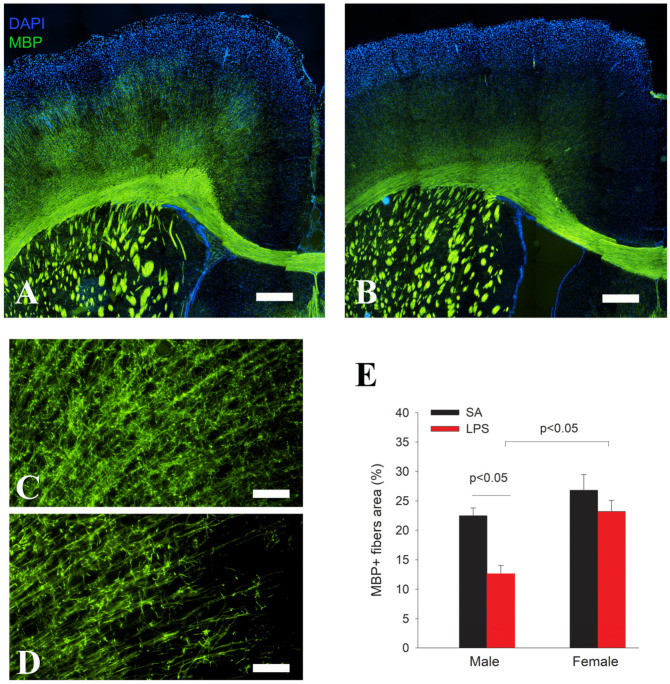
Impaired cortical myelination in LPS-exposed rats at P21. MBP immunofluorescence staining shows a similar pattern and intensity in the major white matter tracks between LPS (**B**) and control rats (**A**), while MBP immunostained fiber density in the cingulate cortex was much lower in the LPS (**D**) group than the control (**C**). Quantification of MBP+ nerve fibers in the cingulate cortex by area fraction shows a significant decrease in myelination in the male but not female rats (**E**). *n* = 10 (5 males, 5 females) in each group. Scale bars: (**A**,**B**) (500 µm), (**C**,**D**) (50 µm).

**Figure 6 brainsci-14-00976-f006:**
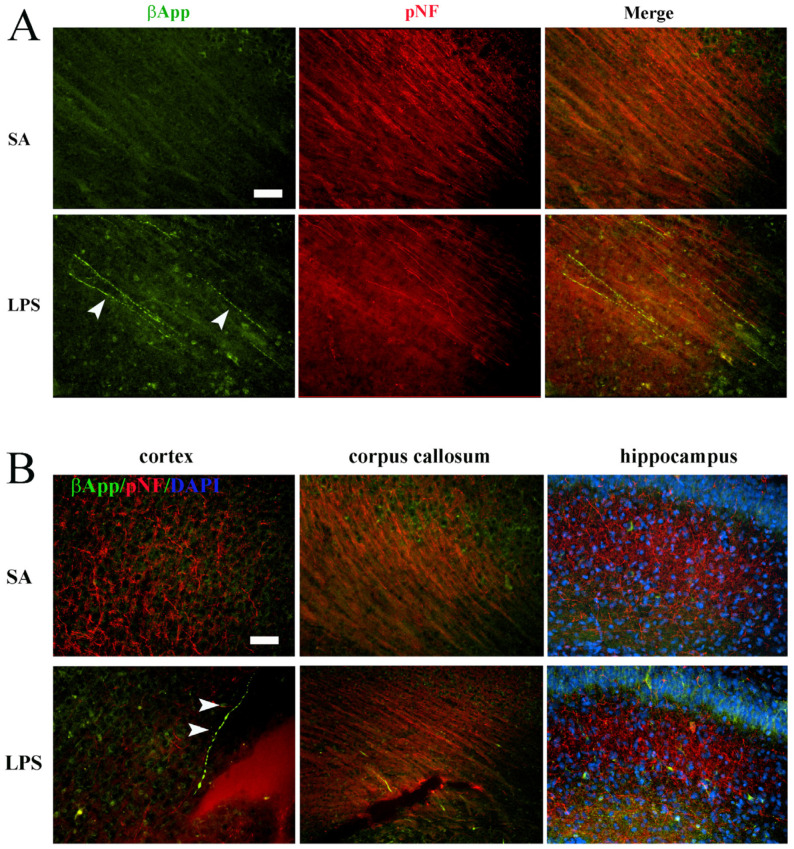
Axonal injury following LPS exposure at P7. (**A**) Double-immunostaining revealed numerous βApp+/pNF+ axons (arrowheads) in the periventricular white matter of the LPS exposed rats. No βApp+/pNF+ axons were observed in the control rats. (**B**) Representative micrographs depict βApp+/pNF+ axons in both the gray matter (cortex and hippocampus) and white matter (corpus callosum). Some of the βApp+ axons exhibit classical “beaded” morphology of axonal degeneration (arrowheads in (**B**)). Scale bars: 50 µm.

**Figure 7 brainsci-14-00976-f007:**
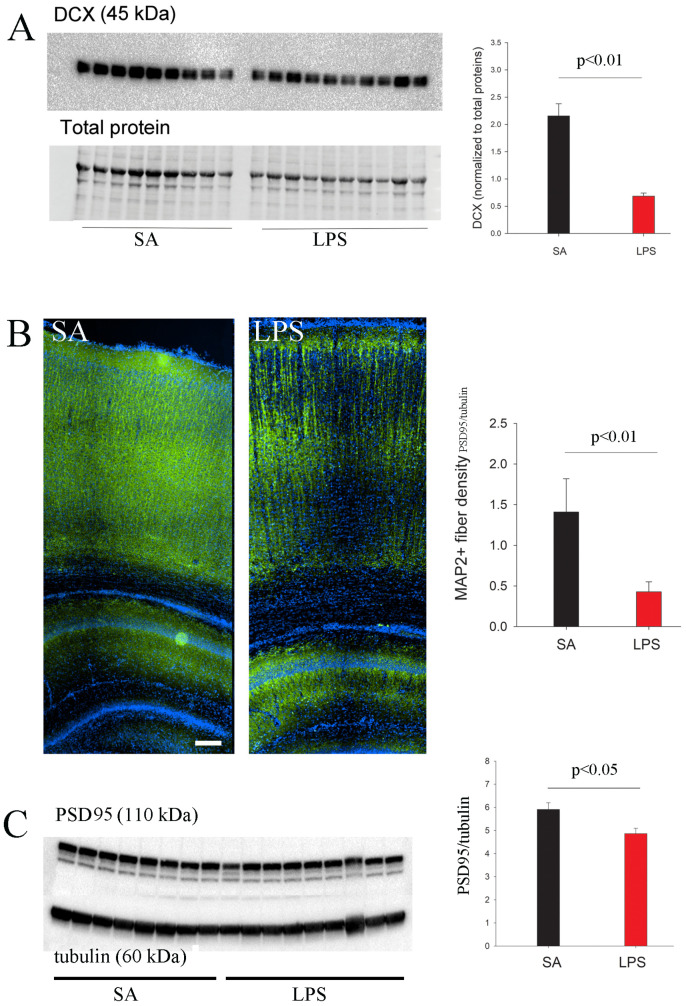
Reduced expression of markers for neurogenesis and synaptic maturation in LPS-exposed rats. (**A**) LPS-mediated neuroinflammation significantly reduced DCX expression in the hippocampus of P7 rats, as determined by Western blot. (**B**) LPS-exposed P7 rats showed lower MAP2+ dendritic density in the cortex. (**C**) A significant decrease in hippocampal PSD95 expression was observed at P21. *n* = 9 in the SA, and *n* = 10 in the LPS group. Scale bar: 100 µm.

**Figure 8 brainsci-14-00976-f008:**
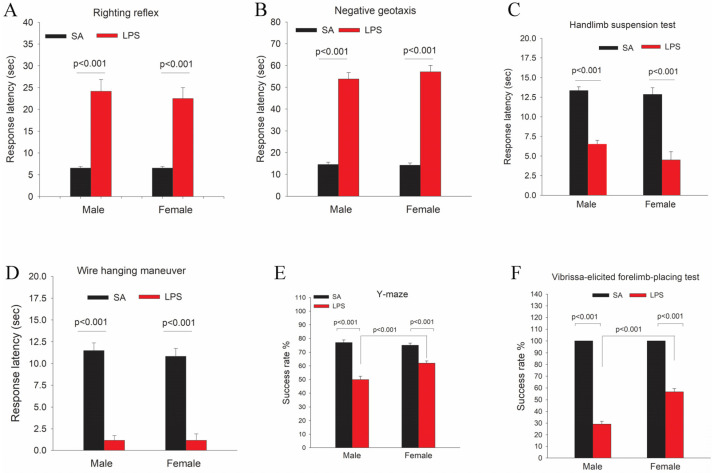
LPS exposure leads to functional impairments as determined by a battery of neurobehavioral tests. LPS-exposed rats showed various behavioral deficits as determined by the righting reflex (**A**), negative geotaxis (**B**), hindlimb suspension (**C**), wire hanging maneuver (**D**), Y-maze (**E**), and vibrissa-elicited forelimb-placing tests (**F**). In addition, LPS-exposed male rats performed much worse than female rats during Y-maze and vibrissa-elicited forelimb-placing tests. *n* = 12 each group (6 males and 6 females).

## Data Availability

Data are not publicly accessible due to privacy concerns. Data supporting the findings in this study can be obtained from the corresponding author upon reasonable request.

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
