# Peer review of "Early Postnatal Neuroinflammation Produces Key Features of Diffuse Brain White Matter Injury in Rats"

_brainsci, 2024, doi:10.3390/brainsci14100976_

Round 1

Reviewer 1 Report

Comments and Suggestions for Authors

This manuscript investigates the impact of early postnatal neuroinflammation on brain development, focusing on diffuse white matter injury (dWMI) in neonatal rats. The research reveals that neuroinflammation induced by intracerebral administration of lipopolysaccharide (LPS) at postnatal day 5 leads to significant neuropathological changes. These include robust activation of microglia and astrocytes, impaired maturation of oligodendrocytes, and myelination deficits. Additionally, LPS exposure results in acute axonal injury and impaired neurogenesis, particularly in the hippocampus, which may underlie observed neurobehavioral deficits. Behavioral assessments demonstrate that LPS-exposed rats exhibit sensorimotor, neuromuscular, and cognitive impairments. The findings highlight the long-lasting effects of early neuroinflammation on brain development, particularly in relation to dWMI, and underscore the need for further research to explore potential therapeutic strategies for mitigating these adverse outcomes.

There are some minor problems in the manuscript.

1. Figure 3 is too small and lacks sufficient clarity. The details are hard to discern, which could make it difficult for readers to interpret the data accurately. It would be beneficial to provide a larger, higher-resolution version of this figure to improve readability.

2. In Figure 4, the blue staining is not clearly identified as either Olig2 or DAPI. This ambiguity can lead to confusion, as it is crucial for understanding what the staining represents in the context of the experiment.

3. The manuscript does not sufficiently address whether PDGFR alpha positivity can definitively represent oligodendrocyte precursor cells (OPCs). This is an important point that needs to be clarified, as PDGFR alpha is a marker often associated with OPCs, but its expression alone may not be conclusive evidence of OPC identity. Further discussion is needed.

Comments on the Quality of English Language

 There are some grammatical errors in the article, such as

Line 38, “one of the well-defined contributory factor”;

Line 49-50, “remain” vs “was”;

Line 52, “might the major”;

Line 61, “impact” vs “causing”;

Line 266, “does not causing”;

Line 268, “significant”;

Line 319, “3-folds” and Line 376, “10-folds”;

Line 409, “marker DCX and NCAM”.

Author Response

Reviewer 1:

Comments:

This manuscript investigates the impact of early postnatal neuroinflammation on brain development, focusing on diffuse white matter injury (dWMI) in neonatal rats. The research reveals that neuroinflammation induced by intracerebral administration of lipopolysaccharide (LPS) at postnatal day 5 leads to significant neuropathological changes. These include robust activation of microglia and astrocytes, impaired maturation of oligodendrocytes, and myelination deficits. Additionally, LPS exposure results in acute axonal injury and impaired neurogenesis, particularly in the hippocampus, which may underlie observed neurobehavioral deficits. Behavioral assessments demonstrate that LPS-exposed rats exhibit sensorimotor, neuromuscular, and cognitive impairments. The findings highlight the long-lasting effects of early neuroinflammation on brain development, particularly in relation to dWMI, and underscore the need for further research to explore potential therapeutic strategies for mitigating these adverse outcomes.

Response: We appreciate the positive comments that recognize the importance of this original work in the field of dWMI.

Minor problems in the manuscript.

Comment1. Figure 3 is too small and lacks sufficient clarity. The details are hard to discern, which could make it difficult for readers to interpret the data accurately. It would be beneficial to provide a larger, higher-resolution version of this figure to improve readability.

Response: A high resolution Figure 3 is provided in the manuscript. To help reviewing and future production, all figures in its original, high resolution formats are submitted as individual file.  

Comment 2. In Figure 4, the blue staining is not clearly identified as either Olig2 or DAPI. This ambiguity can lead to confusion, as it is crucial for understanding what the staining represents in the context of the experiment.

Response: Thank you for clarifying this issue. DAPI is labeled in figure 4A.

Comment 3. The manuscript does not sufficiently address whether PDGFR alpha positivity can definitively represent oligodendrocyte precursor cells (OPCs). This is an important point that needs to be clarified, as PDGFR alpha is a marker often associated with OPCs, but its expression alone may not be conclusive evidence of OPC identity. Further discussion is needed.

Response: We agree with the concern. A brief discussion was provided to clarify this issue (line 370-373).

Comment 4: Quality of English Language

 There are some grammatical errors in the article, such as

Line 38, “one of the well-defined contributory factor”;

Line 49-50, “remain” vs “was”;

Line 52, “might the major”;

Line 61, “impact” vs “causing”;

Line 266, “does not causing”;

Line 268, “significant”;

Line 319, “3-folds” and Line 376, “10-folds”;

Line 409, “marker DCX and NCAM”.

Responses: Thank you very much for your careful review. We paid close attention to the language and corrected grammatical errors, including these mentioned above, during the revision.

Reviewer 2 Report

Comments and Suggestions for Authors

The study aims to investigate the impact of LPS-induced neuroinflammation on neurobiological and behavioral outcomes in rats between P7 and P21. The authors discovered that injecting LPS into the brain during the early postnatal period caused significant activation of microglia and astrocytes, as well as deficits in myelination. 

Overall, this study demonstrates that neuroinflammation in the early postnatal period contributes to the pathophysiological features of diffuse white matter injury.  However, there are still gaps in the paper and how the data has been presented.

Please ensure that the timeline for the experiments is included in the methods section. Additionally, it would be helpful to know why the authors chose to display the IHC data from P7 and not P21. Is it assumed that at P21, LPS-treated rats would exhibit impairments in OL maturation and myelination? Moreover, the authors should address the gender-specific impaired cortical myelination observed in P21 rats. The authors have not provided an explanation or discussion for this. It's important to note that a reduction in DCX expression does not necessarily indicate reduced or impaired neurogenesis. DCX is a cytoskeleton protein that aids in the migration of newborn neurons. Consequently, the authors should consider conducting additional experiments to support such claims. It would be beneficial for the authors to present IHCs of the hippocampus with DCX to illustrate differences in neurogenesis. Lastly, please ensure that individual data points are shown on each graph.

Comments on the Quality of English Language

The manuscript is well-written. However, there are a few minor grammatical mistakes.

Author Response

Reviewer 2

Comments

The study aims to investigate the impact of LPS-induced neuroinflammation on neurobiological and behavioral outcomes in rats between P7 and P21. The authors discovered that injecting LPS into the brain during the early postnatal period caused significant activation of microglia and astrocytes, as well as deficits in myelination.

Overall, this study demonstrates that neuroinflammation in the early postnatal period contributes to the pathophysiological features of diffuse white matter injury.  However, there are still gaps in the paper and how the data has been presented.

  1. Comments: Please ensure that the timeline for the experiments is included in the methods section.

Response: a timeline graph is included in the Material and Methods section (line 97)

  1. Comments: Additionally, it would be helpful to know why the authors chose to display the IHC data from P7 and not P21. Is it assumed that at P21, LPS-treated rats would exhibit impairments in OL maturation and myelination?

Response: Most of the IHC studies were performed at P7, except myelination (MBP) was analyzed at P21 (Figure 4). There are several considerations for choosing the specific timeline for specific markers: 1. In rat, myelination starts around P10 and is abundant at P21, therefore we choose P7 for assessing markers of OPC and immature OLs. 2. We are interested in assessing acute effect of neuroinflammation on OL lineage and neural processes (axonal, dendritic, and synaptic markers), so we chose P7 (48 h post LPS challenge) instead of P21. Additionally, some of those markers (e.g. MAP2) significantly downregulated at P21 as compared to P7.

  1. Comments: Moreover, the authors should address the gender-specific impaired cortical myelination observed in P21 rats. The authors have not provided an explanation or discussion for this.

Response: Thank you for pointing this out, this is a great question. We did mention that data from male and female rats were analyzed separately, and if no sex difference were detected, data were compiled into a single group. To further address sex-difference in myelination observed in P21, we briefly discussed this issue (line 466-469)

  1. Comments: It's important to note that a reduction in DCX expression does not necessarily indicate reduced or impaired neurogenesis. DCX is a cytoskeleton protein that aids in the migration of newborn neurons. Consequently, the authors should consider conducting additional experiments to support such claims. It would be beneficial for the authors to present IHCs of the hippocampus with DCX to illustrate differences in neurogenesis. Lastly, please ensure that individual data points are shown on each graph.

Response: We agree with the comment that reduction in DCX does not necessarily indicate reduced or impaired neurogenesis. A brief discussion is included in the revision (Line 408-414). We performed DCX IHC; however, we found it difficult to accurately quantify DCX+ cells because a significant portion of DCX immunostaining is located outside cells. As an alternative, we performed ki67 IHC to indicate changes of neurogenesis, since Ki67+ cells in the postnatal SVZ and dentate gyrus (DG) are considered as neural stem cells. We found that Ki67+ cells were significantly reduced in the DG but not SVZ (supplement figure), suggesting that neuroinflammation might indeed suppressed neurogenesis. For the graph, the journal does not require specific data format, so we keep the current bar graph format.

Comments on the Quality of English Language

The manuscript is well-written. However, there are a few minor grammatical mistakes.

Response: we tried our best to correct grammatical mistakes during revision.